# Reporting quality of published reviews of commercial and publicly available mobile health apps (mHealth app reviews): a scoping review protocol

Norina Gasteiger [ID],[1,2] Gill Norman [ID],[1,2,3] Rebecca Grainger [ID],[4]
Charlotte Eost-Telling [ID],[1,2] Debra Jones,[1] Syed Mustafa Ali,[2,5]
Sabine N van der Veer,[2,5] Claire R Ford,[1] Alex Hall,[1] Kate Law,[1,6] Matthew Byerly,[7]
Alan Davies [ID],[5] Deborah Paripoorani,[2,8] Chunhu Shi [ID],[1,2] Dawn Dowding[1,2]

For numbered affiliations see end of article.

**Correspondence to**
Dr Norina Gasteiger;
norina.gasteiger@manchester.ac.uk

## ABSTRACT

**Introduction** Reviews of commercial and publicly available smartphone (mobile) health applications (mHealth app reviews) are being undertaken and published. However, there is variation in the conduct and reporting of mHealth app reviews, with no existing reporting guidelines. Building on the Preferred Reporting Items for Systematic Reviews and Meta-Analyses (PRISMA) guidelines, we aim to develop the Consensus for APP Review Reporting Items (CAPPRRI) guidance, to support the conduct and reporting of mHealth app reviews. This scoping review of published mHealth app reviews will explore their alignment, deviation, and modification to the PRISMA 2020 items for systematic reviews and identify a list of possible items to include in CAPPRRI.

**Method and analysis** We are following the Joanna Briggs Institute approach and Arksey and O'Malley's five-step process. Patient and public contributors, mHealth app review, digital health research and evidence synthesis experts, healthcare professionals and a specialist librarian gave feedback on the methods. We will search SCOPUS, CINAHL Plus, AMED, EMBASE, Medline, APA PsycINFO and the ACM Digital Library for articles reporting mHealth app reviews and use a two-step screening process to identify eligible articles. Information on whether the authors have reported, or how they have modified the PRISMA 2020 items in their reporting, will be extracted. Data extraction will also include the article characteristics, protocol and registration information, review question frameworks used, information about the search and screening process, how apps have been evaluated and evidence of stakeholder engagement. This will be analysed using a content synthesis approach and presented using descriptive statistics and summaries. This protocol is registered on OSF (https://osf.io/5ahjx).

**Ethics and dissemination** Ethical approval is not required. The findings will be disseminated through peer-reviewed journal publications (shared on our project website and on the EQUATOR Network website where the CAPPRRI guidance has been registered as under development), conference presentations and blog and social media posts in lay language.

## STRENGTHS AND LIMITATIONS OF THIS STUDY

⇒ This review will be conducted systematically, with data extraction informed by the Preferred Reporting Items for Systematic Reviews and Meta-Analyses 2020 reporting items and a previous scoping review.

⇒ The protocol has had input from a multidisciplinary team of mHealth app review, digital health and evidence synthesis experts, healthcare professionals, a librarian, and patient and public contributors.

⇒ The broad scope of health topics to be included in the review increases the generalisability of the findings.

⇒ In line with scoping review guidance, a quality appraisal of the included studies will not be conducted.

⇒ Only mHealth app reviews reported in English will be included, meaning some other relevant reviews published in other languages may not be included.

## INTRODUCTION

In 2021, it was recorded that there are more than 350 000 health applications (apps).[1] These applications are increasingly being integrated into healthcare; supporting professionals in their clinical practice[2] and empowering patients to manage and monitor their health conditions.[3 4] However, the quality and reliability of mobile health (mHealth) apps vary significantly,[5] as developers can release smartphone health apps without any evaluation, meaning it is a challenge for health professionals and users without expertise in health research and digital technology to identify and evaluate the suitability of mHealth apps for their use.

This has led to the emergence of a new method: systematic reviews of commercial and publicly available mHealth apps (hereafter called mHealth app reviews). This provides a standard approach to identify mHealth apps relevant to a particular use case

BMJ

and assess aspects such as quality and functionality. Many mHealth app reviews have now been published, for varied topics including genetics,[6 7] patient-reported outcomes in oncology,[8] mental health,[9 10] rheumatoid arthritis,[11] strength training,[12] menopause,[13] exercise,[14 15] hand hygiene,[16] atrial fibrillation,[17] pain[18–20] and smoking cessation.[21 22] These reviews can serve as a valuable resource for healthcare decision-makers, practitioners, patients and the general public seeking high-quality mHealth apps; can identify gaps in the field and may guide researchers and industry in developing new products.

While mHealth app reviews share features with tradicial systematic literature reviews, they differ substantially in their methods and reporting[23] due to the review being of commercial and publicly available products on app stores, instead of published literature. Examples of traditional systematic reviews of literature describing apps include those on monitoring and managing mental health symptoms[24] and self-managing pregnancy.[25] While literature reviews can tell us about the effectiveness of apps which have been evaluated and the results published, they do not provide a comprehensive overview of all apps that are commercially or publicly available for use by patients, healthcare professionals and the public. mHealth app reviews also differ as there are no formal requirements for the protocol to be registered, searches take place on app libraries, screening often takes place on Excel (rather than using specifically designed tools like Rayyan[26] or Covidence[27]), and they are more challenging to replicate, as apps may emerge, disappear or be updated between searches.[23]

The EQUATOR Network[28] provides an array of guidelines for reporting evaluations of digital technologies, such as the CONSORT-EHEALTH checklist (an extension of the CONSORT checklist tailored for reporting randomised controlled trials of web-based and mobile health interventions[29]), and guidance on reporting evaluations of specific technologies, such as sensors,[30] mHealth interventions,[31] telehealth in clinical trials[32] and smartphone-delivered ecological momentary assessments.[33] There are also several extensions of the Preferred Reporting Items for Systematic Reviews and Meta-Analyses (PRISMA) reporting guidelines[34] available for different types of literature reviews.

In contrast, no reporting guidelines exist for mHealth app reviews, and we are not aware of any currently in development. The need for a reporting standard for health app-focused reviews was emphasised in a scoping review published in 2020.[35] The authors reviewed 26 mHealth app reviews published between 2012 and 2018 and found issues in their reporting. For example, the date of the searches was sometimes unclear (38%, 10/26) or absent (15%, 4/26). The number of reviewers involved was also poorly reported in 58% of reviews, and in 83%, it was unclear whether screening was performed independently. Another important finding was the absence of clinical recommendations or reporting on clinical efficacy, found in 77% of the included reviews. Adhering to

reporting guidelines may help to minimise the presence of these inconsistencies in reporting and ensure that standardised information is presented.

If a key purpose of an mHealth app review is for people to be able to identify the best product for a particular purpose, it is important to further explore how app review authors have evaluated the apps and reported on their outcomes. For example, some previous app reviews have considered the accessibility of the apps by generating readability metrics on the written content.[6 8 13] This is especially important for patient or public-facing apps. There are also various approaches to reporting on efficacy (ie, that the app results in intended outcomes). This may include searching for evidence of a previous evaluation within the app itself, on developers' websites, or for published literature or conference abstracts in academic databases. For example, the Mobile App Rating Scale (MARS)[36] is commonly used in mHealth app reviews and has an item which addresses whether an app has been trialled or tested. Previous examples of app reviews have approached this item by excluding the item entirely[11] or searching Google Scholar.[6 8 13 16 18] It is unclear how other authors have evaluated the apps and how this has informed their recommendations of apps as being high quality. Understanding the nature of the evaluations (especially efficacy evidence) seems essential in mHealth app reviews as readers, including health and care workers, patients and the public, and healthcare decision-makers may use these to choose which apps to use.

We have previously discussed the methodological considerations for conducting systematic mHealth app reviews; introducing a seven-step method and the TECH framework (Target user, Evaluation focus, Connectedness and the Health domain) for developing research (review) questions and determining app eligibility criteria.[23] This is the first stage of a broader project that aims to systematise the process of conducting and reporting mHealth app reviews.

The next step is to develop reporting guidelines to support authors of mHealth app reviews in transparently presenting their methods and findings. The field of app reviews is rapidly developing and expanding so a new scoping review is required to update that previously reported, which had a search date of 2018.[35] A preliminary search of the Cochrane Database of Systematic Reviews, Google Scholar and Joanna Briggs Institute (JBI) Evidence Synthesis was conducted, and no current or ongoing systematic reviews or scoping reviews on the topic were identified. Therefore, we are undertaking a scoping review to build on and update the scoping review by Grainger et al.[35]

## OBJECTIVES

In line with guidance for developing reporting guidelines,[37] the next step is to search for relevant evidence on the quality of the reporting of published mHealth app reviews. The aim of this work is, therefore, to conduct

a scoping review on published mHealth app reviews to explore their alignment, deviation and modification to the PRISMA 2020 items and identify a list of possible items to include in the new Consensus for APP Review Reporting Items (CAPPRRI) guideline.

## METHODS
### Scoping review
The methods for this scoping review were developed in alignment with the JBI approach for scoping reviews[38] and reviewed by a group of patient and public contributors and an advisory group consisting of mHealth app review, digital health research and evidence synthesis experts and National Health Service (NHS) healthcare professionals interested in app reviews. The review will be carried out using the five-step process for conducting scoping reviews, originally outlined by Arksey and O'Malley.[39] This protocol has already been registered and made publicly available on OSF (https://osf.io/5ahjx). The final review will be reported using the PRISMA extension guidelines for Scoping Reviews.[40] This protocol has been reported using the PRISMA-P extension[41] (see online supplemental appendix 1). We will start the review on 2 January 2024 and complete it by 2 September 2024.

### Procedures
#### Identifying the initial research question
We used the Study, Data, Methods and Outcomes (SDMO) acronym to inform our research questions and eligibility criteria, which has been recommended when conducting reviews on methodology or theory.[42] Additionally, as suggested by Levac et al,[43] we considered the purpose and expected outputs of the review to assist in writing the research questions. The purpose and expected outputs are primarily a list of potential reporting items used to inform the future CAPPRRI guideline. Building on the PRISMA 2020[34] items is appropriate as many app review authors already informally use the PRISMA items to report their work or have attempted to amend the PRISMA flow chart when reporting their app search and screening process.[6 8 11–13 16–18 21]

The second question seeks to understand what outcomes were evaluated in mHealth app reviews (eg, usability, functionality, privacy, accessibility and efficacy). This builds on the previous review,[35] which found that most of the app reviews did not make clinical recommendations or report clinical efficacy (ie, whether the app could meet desired outcomes in a clinical context). We are, therefore, interested in understanding what the outcomes were in general, and whether any of the app reviews reported on efficacy in any other sense such as satisfaction, increased knowledge or perceived support.

The two key questions are as follows:
1. In published reviews of commercial and publicly available mHealth apps how does reporting align with or deviate from the PRISMA 2020 items? Have authors

**Table 1** Keywords used to identify literature

| Technology | Review type |
|---|---|
| ► Mobile<br>► Smartphone<br>► "Cell phone"<br>► mHealth<br>► "mobile health"<br>► eHealth<br>► Tele* (to include telehealth, telemedicine and other variations) | (App or apps or application or applications) adj5* review |

*The exact wording depends on the database and may include adj5, W/5 or N5. The ACM Digital Library does not allow the proximity function so 'app* review' will be used instead.

used items that directly align with PRISMA 2020 items or have these been modified?
2. What outcomes did the mHealth app reviews evaluate and report on?

### Identifying relevant studies
We will search the SCOPUS, CINAHL Plus (via EBSCO), AMED (via Ovid), EMBASE (via Ovid), Medline (via Ovid), APA PsycINFO and ACM Digital Library databases for published mHealth app reviews, under the guidance of a teaching and learning librarian who has given input on the search strategy. Reference lists of eligible articles will also be handsearched for additional sources (snowballing) while a forward citation approach will be used to identify app reviews that have cited earlier published work.

The key terms used to build the search strategy are shown in table 1, with the full search strategy presented in online supplemental appendix 2. Where appropriate, subject headings will be applied to the databases. The keywords will be separated by the 'OR' Boolean operator. The technology and review type keywords will be separated by the 'AND' Boolean operator. The proximity function using five words will be used for the review type keywords, to include different variations of app review, such as review of apps, review of smartphone apps or review of patient-facing mobile health applications.

Publication date will be limited from 1 January 2007, as the first iPhone (and first smartphone) was introduced on 29 June 2007 so there will be no apps available to review before 2007.

### Study selection
Table 2 presents the eligibility criteria for the literature, using the SDMO acronym.[42] Broadly, our inclusion criteria are as follows:
► Reviews of commercial and publicly available mobile (smartphone) apps published in English and after 1 January 2007 (types of study).
► that have a health focus (type of data),
► include any method and measure of evaluating apps (eg, MARS or user ratings) and,

**Table 2** Eligibility criteria for the mHealth app reviews to be included in the scoping review

| SDMO | Inclusion criteria | Exclusion criteria | No limits |
|---|---|---|---|
| Types of study | Reviews of commercial and publicly available mobile apps<br>► Must be focused on smartphone (mobile) apps.<br>► Can be identified as systematic, scoping or without a specific approach named.<br>► Some app reviews may be combined with other literature reviews or reviews of other apps. These will only be included if detail is reported separately on the smartphone app review methods and results.<br>► Reviews including other technology (eg, iPads, digital assistants, virtual reality headsets or smartwatches) will only be included if the focus is on smartphone apps and the other technology is used only to operationalise some of the functions.<br>English language<br>Published on or after 1 January, 2007 | Literature reviews<br>Reviews of other technology or apps (eg, websites, computer apps, iPad apps)<br>Full text not available<br>► Exclude abstracts and documents where there is insufficient information or the full text is not available.<br>Not in English<br>Published before 1 January 2007 | Document type<br>► Any document type will be included if there is a full text available so that enough information can be extracted (eg, full-length conference papers, journal articles, book chapters).<br>Smartphone device<br>Operating system requirements<br>App markets<br>Geography (location) |
| Types of data | Health focus:<br>Must be focused on a health topic, whereby the apps are marketed for physical or mental health or general well-being. This may include (but is not limited to) apps that educate, empower or inform users on a health topic (eg, genetics), self-monitor/manage or change health behaviours (eg, sleep, nutrition, exercise or smoking cessation) or are used for social support or in health systems or by patients, administrators and health and care workers or decision-makers (eg, screening, diagnosis, triage, appointment-booking, remote monitoring, decision-making, training and treatment). | Not focused on health topics | Health topic<br>► Apps can be for any health topic.<br>Intended users of apps<br>► Apps can be for any stakeholder, including patients, the public, health professionals and the health system. |
| Types of methods | Any method (and measure) of evaluating apps can be included, such as using validated measures (eg, MARS[36]), synthesising content presented within the app or user ratings and reviews on app markets. | N/A | App evaluation measures and methods. |
| Types of outcomes | Any outcome, including those related to evaluating app quality, functionality, privacy and security, accessibility or efficacy. This would also include app reviews that simply focus on identifying which apps were available, summarised their content or described the extent to which they adhere to best clinical practice/guidelines. | N/A | Any outcomes. |

MARS, Mobile App Rating Scale; N/A, not available; SDMO, Study, Data, Methods and Outcomes.

► have any outcome or focus, including focusing on the availability of apps or evaluation (eg, quality, functionality, privacy and security or adherence to clinical guidelines).

We will only include mHealth app reviews published in English. However, we will separately list articles that were excluded due to language, which can enable others to easily identify these papers for subsequent reviews.

The final search results will be imported into Rayyan[26] for deduplication and screening. As suggested by Levac *et al*,[43] the screening process will be iterative and use a team-based approach. An initial meeting will be held with all researchers involved in the screening process to discuss interpretation of the eligibility criteria and reach a shared understanding after an initial set of records have been pilot screened. A two-step process will then be followed. First, two researchers will independently review each abstract/title against the eligibility criteria. Second, the full text of records potentially eligible based on abstract/title screening will be reviewed by two researchers independently. A meeting will be held after the second stage to discuss and reach consensus where there is disagreement. A third reviewer will make the final decision if consensus cannot be reached. Depending on the number

**Table 3** Information to be extracted from the articles

**Article characteristics**

| | |
|---|---|
| Title | ▶ Title of the article. <br> ▶ Does it name a review method in the title (eg, systematic, scoping, app review)? |
| Date | ▶ Year of publication. |
| Journal | ▶ Name of the journal the app review is published in. <br> ▶ Name and contact information of the editor-in-chief.* |
| Authors | ▶ Name and contact information of all authors.* <br> ▶ Source of funding (if any). |
| Objective | ▶ Aim or research question. |
| Topic and context | ▶ The health problem being explored. <br> ▶ App target user. <br> ▶ Context: where the app is to be used, including location of care (acute, primary healthcare, community, long-term care, etc) and geographical location. |

**Protocol and registration**

| | |
|---|---|
| A priori review registration | ▶ Was the study protocol registered and was the protocol available?† |

**Review question**

| | |
|---|---|
| Review question frameworks | ▶ Was a framework used to write the review question (eg, PICO)? If so, which? <br> ▶ Did this align with any of the TECH components? If so, which? |

**Reporting guidelines**

| | |
|---|---|
| Reporting guidelines | ▶ Did authors state which reporting guideline they used? If so, which? <br> ▶ Did the authors clearly mention amending any guidelines? |
| Alignment, deviation and modification to the PRISMA (2020) items (see online supplemental appendix 3) | ▶ For each of the PRISMA (2020) items, identify whether the information was reported as is, or how this was modified (if applicable). |

**Search and screening**

| | |
|---|---|
| Flow charts/diagrams | ▶ Did the authors present a flow diagram for the search and screening process? <br> ▶ Did the authors report a PRISMA (2020) flow chart for new or updated systematic reviews (see online supplemental appendices 4 and 5)? <br> ▶ Were amendments made to the PRISMA (2020) flow charts (see online supplemental appendices 4 and 5)? If yes, which? |
| App store search | ▶ Were the apps store(s) searched described including the (a) keywords, (b) countries/location and (c) a clear description of dates of the search?† <br> ▶ Was the method for identifying and removing duplicate apps clearly described, including metadata (eg, version numbers) used to determine if apps were duplicates?† <br> ▶ Were the number and independence of people screening apps described?† <br> ▶ Were limits on inclusion of apps based on other factors clearly described (lite or full version, paid or free versions, non-English)?† <br> ▶ Where the same app featured in different app stores (multiplatform apps), was there a clear statement of which apps were included?† <br> ▶ When apps were downloaded to phones for data extraction, were the phone model and version of the operating system clearly reported?† |

**Evaluating the apps and making recommendations**

| | |
|---|---|
| Outcomes | ▶ Which outcomes did the review evaluate? For example, quality, functionality, usability, privacy, efficacy, accessibility or other |
| Quality, functionality, usability and other assessments | ▶ Was a best practice content tool (eg, clinical guideline) used to evaluate app content or quality? Was the source identified, were any modifications made described and was the use of the guideline justified?† <br> ▶ Was an app quality, functionality and/or usability evaluation undertaken using established measures and frameworks (eg, MARS)? <br> ▶ Are any other tools/instruments used to evaluate app quality and/or usability and was the source of these described?† <br> ▶ Were security and privacy considered? If so, what aspects were included? For example, login, password, privacy policy, access to microphone or camera, encryption or data sharing. |

Continued

**Table 3** Continued

| | |
|---|---|
| Efficacy | ▶ Did the authors report on the efficacy of the apps?<br>▶ If yes, where did they obtain this information and what methods were used? |
| Accessibility | ▶ Was the accessibility of the apps evaluated, considering the different needs of target users? |
| Recommendations | ▶ Were any apps recommended for use in the clinical setting or by people with chronic health conditions?†<br>▶ Were any apps recommended overall? What informed this recommendation? |
| **Stakeholder engagement or consultation** | |
| Patient, public and expert engagement | ▶ Were any patients or members of the public involved in the app review?<br>▶ Were any other stakeholders involved or consulted (eg, industry partners, clinicians or software developers)? If so, which?<br>▶ Was a lay summary provided? If so, were any apps clearly recommended? |
| **Other** | |
| Other domains | ▶ Any other domains reported on? If yes, what and how? |

*This information will not be published publicly as part of the review. It will be used to create a database of authors of mHealth app reviews and journals who have published them. This is required for the next step of the project- the Delphi (consensus-building) study, as mHealth experts will be participants and potential partners in this process.
†These data extraction items were taken from the scoping review conducted by Grainger *et al*.[35]
PRISMA, Preferred Reporting Items for Systematic Reviews and Meta-Analyses.

of articles, two teams of two researchers may perform the screening process, with a fifth researcher available to resolve disagreements. The search and screening process will be presented as a flow chart.

### Charting the data
A data extraction sheet will be created in Excel using headers related to our seven-step method and TECH framework,[23] the PRISMA 2020 items,[34] whether and how authors modified them, additional information reported, and methods used to appraise the apps' quality, functionality or efficacy. Some items have also been taken from the review conducted by Grainger *et al*[35] as these capture details unique to app reviews (eg, how app stores were searched). Table 3 presents the data extraction items.

Consistent with recommendations by Levac *et al*,[43] we will take an iterative approach to charting by continually updating the data-charting form as needed. The research team will first pilot the data extraction sheet, by extracting the data from one app review, with a discussion afterwards to ensure consistency in interpretation of the items. Data will then be extracted from the other articles, with one author extracting the information and another checking this. Depending on the final number of included articles, this will be split between the researchers.

It has been suggested that some scoping reviews should also include quality assessments of the methodology used in the articles.[43] However, as this is not the focus of our review and as no specific quality assessment tool currently exists for mHealth app reviews, the quality of the included studies will not be assessed.

### Collating, summarising and reporting the results
Similar to the previous review[35] and as recommended by Arksey and O'Malley,[39] we will report data as frequencies (where possible) to determine which items were reported as is, or whether they were modified.

Information that cannot be reported as frequencies, on how the PRISMA 2020 items were modified and how other relevant information was reported will be summarised using a content synthesis approach, to help identify new items for the CAPPRRI guideline.

The results overall will be reported using descriptions and examples while some of the numerical results will also be presented using tables and figures.

### Strengths and limitations
This scoping review will be conducted in a systematic and rigorous manner, with data extraction informed by the existing PRISMA 2020[34] reporting items and a previous scoping review.[35] It also adheres to existing guidance on conducting scoping reviews, including from the JBI[38] and Arksey and O'Malley[39] and has had input from a multi-disciplinary team of mHealth app review, digital health and evidence synthesis experts, NHS healthcare professionals, a librarian and patient and public contributors. The breadth of its scope of health topics (and methodological focus) also means that the findings will be widely generalisable.

A fundamental limitation is the inability to assess the quality of the included reviews, due to an absence of quality appraisal tools for reviews of commercial and publicly available mHealth apps. This limitation means that low-quality app reviews may also contribute to the development of the future CAPPRRI guideline. However, we will mitigate this in the next phase of the project, in which a Delphi study with experts will help to prioritise the items. Another limitation concerns restricting the included mHealth app reviews to those reported in English which may lead to other relevant reviews being excluded.

## Patient and public involvement

We have established a Patient and Public Involvement and Engagement (PPIE) group to give feedback on our project. The group has been consulted to provide input on the protocol and suggested additional items that should be extracted (ie, whether the accessibility of the apps was evaluated and whether a lay summary was provided). They also gave ideas for how the findings should be disseminated. We will continue to consult with them throughout the review; this will inform the iterative aspects of the scoping review process and will help to guide the findings and their dissemination.

## ETHICS AND DISSEMINATION

Ethical approval is not required to conduct this scoping review which will use only previously published data.

The findings of this scoping review will be disseminated through peer-reviewed journal publications which will be shared on our project website and on the EQUATOR Network website where the CAPPRRI guideline has been registered as under development, in addition to conference presentations and blog posts and short summaries in lay language on professional social media.

## CONCLUSION

This protocol describes how we will conduct a scoping review on published mHealth app reviews to explore their alignment, deviation and modification to the PRISMA 2020 items and identify a list of possible items to include in the new CAPPRRI reporting guideline. The results will inform the next phase in developing the CAPPRRI guideline: a Delphi study to reach a consensus on which items are most relevant and important to include in the guideline.

**Author affiliations**
[1]Division of Nursing, Midwifery & Social Work, The University of Manchester, Manchester, UK
[2]NIHR Applied Research Collaboration Greater Manchester, Manchester, UK
[3]NIHR Innovation Observatory, Population Health Sciences Institute, Newcastle University, Newcastle, UK
[4]Department of Medicine, University of Otago, Wellington, New Zealand
[5]Centre for Health Informatics, Division of Informatics, Imaging & Data Sciences, Manchester Academic Health Science Centre, The University of Manchester, Manchester, UK
[6]The Christie Hospital NHS Trust, Manchester, UK
[7]The University of Kansas School of Medicine, Wichita, Kansas, USA
[8]EMERGING Research Team, Manchester Royal Infirmary, Manchester, UK

**Acknowledgements** We would like to acknowledge Mr Michael Stevenson for assisting with the search strategy. We would also like to thank our patient and public contributors for their input—thank you Amber McAvoy, Eric Lowndes, Ashgan Mahyoub and Beatrice Namu. We acknowledge Mrs Amy Vercell and Dr Lisa McGarrigle who are part of our team.

**Contributors** NG, GN and DD designed the protocol. NG wrote the first draft of the manuscript, with support from GN. NG, GN, RG, CE-T, DJ, SMA, SNvdV, CRF, AH, KL, MB, AD, DP, CS and DD revised the protocol critically and approved the final manuscript.

**Funding** This work is funded by the National Institute for Health and Care Research Applied Research Collaboration Greater Manchester (grant award: NIHR200174).

**ORCID iDs**
Norina Gasteiger http://orcid.org/0000-0001-7801-7417
Gill Norman http://orcid.org/0000-0002-3972-5733
Rebecca Grainger http://orcid.org/0000-0001-9201-8678
Charlotte Eost-Telling http://orcid.org/0000-0002-9568-3195
Alan Davies http://orcid.org/0000-0001-5737-5629
Chunhu Shi http://orcid.org/0000-0003-0151-0451

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
