## [Reviewer comments · BMJ Open]

ARTICLE DETAILS

TITLE (PROVISIONAL)	The reporting quality of published reviews of commercial and publicly available mobile health apps (mHealth app reviews): A scoping review protocol
AUTHORS	Gasteiger, Norina; Norman, Gill; Grainger, Rebecca; East-Telling, Charlotte; Jones, Debra; Ali, Syed; van der Veer, Sabine N.; Ford, Claire; Hall, Alex; Law, Kate; Byerly, Matthew; Davies, Alan; Paripoorani, Deborah; Shi, Chunhu; Dowding, Dawn

VERSION 1 – REVIEW

REVIEWER	Wattanapisit, Apichai School of Medicine, Walailak University
REVIEW RETURNED	18-Jan-2024

GENERAL COMMENTS	Thank you for the opportunity to review this interesting protocol manuscript. The manuscript is well-written, and the design of this scoping review is rigorous and scientifically sound. Given the registered status of the protocol on OSF and its scientific and writing quality, I recommend this manuscript for publication. I have a minor comment for the journal. I noticed some footnotes at the bottom of pages 8 and 10. I believe they should serve as the table legends for Table 1 and Table 3.
---

REVIEWER	Ghaffari, Arash Aalborg Universitetshospital, Interdisciplinary Orthopaedics
REVIEW RETURNED	13-Feb-2024

GENERAL COMMENTS	Dear authors, I appreciate the opportunity to evaluate your manuscript titled "The reporting quality of published reviews of commercial and publicly available mobile health apps (mHealth app reviews): A scoping review protocol." While the study holds promise, I would like to address certain concerns. In my opinion, it would be advantageous for this protocol to be an update of your previous work (reference 28), and I hope these suggestions will help in refining your protocol: 1- The effort to create a PRISMA extension for mHealth app evaluations is admirable. Nevertheless, it is necessary to provide a more explicit explanation for this objective. Clarifying the reasoning, especially considering the already available guidelines for reporting app technologies, would assist readers in comprehending the need for a new reporting guideline. The authors need to clearly explain how their suggested expansion to PRISMA will directly tackle certain deficiencies in the
---

	documentation of mHealth app reviews (and not systematic literature reviews). Furthermore, the protocol should clarify the distinctive obstacles in reporting mHealth app reviews that justify a new expansion, taking into account general issues (mentioned by the authors in the paragraph 5 of the Introduction) that could be resolved without creating a new reporting guideline. In addition, the absence of quality appraisal aligning with the nature of scoping reviews present some challenges for creating an extension for PRISMA guideline. 2- I hold a different viewpoint from the authors when it comes to the broad range of health topics (smoking cessation, mental health, strength training, and ...). I believe that this wide scope can make it difficult to draw specific and actionable conclusions. To overcome this challenge, I suggest adopting a more targeted approach that ensures practical and applicable outcomes. 3- I find it confusing that the authors' aim is to create an extension for PRISMA, which is designed for systematic reviews, and yet they exclude studies literature reviews. If the intention is to develop an extension for reporting systematic reviews, it would be necessary to identify the problems with reporting in literature reviews. 4- In addition, it is important to note that the title states "The reporting quality of ...", while the protocol explains that the review does not actually include assessing quality because there are no specific tools available for reviewing mHealth apps. This potential inconsistency needs to be resolved to create a more coherent presentation. 5- I noticed a significant dependence on the authors' own earlier research in the references. I suggest expanding the review of the literature to encompass a wider array of pertinent studies, guaranteeing a thorough and impartial comprehension of the existing terrain.
--	---

VERSION 1 – AUTHOR RESPONSE

Reviewer: 1. Dr. Apichai Wattanapisit, School of Medicine, Walailak University

Thank you for the opportunity to review this interesting protocol manuscript. The manuscript is well-written, and the design of this scoping review is rigorous and scientifically sound. Given the registered status of the protocol on OSF and its scientific and writing quality, I recommend this manuscript for publication.

Thank you very much for taking the time to review our protocol.

I have a minor comment for the journal. I noticed some footnotes at the bottom of pages 8 and 10. I believe they should serve as the table legends for Table 1 and Table 3.

We have removed the footnotes and replaced them with notes under Tables 1 and 3.

Reviewer: 2. Dr. Arash Ghaffari, Aalborg Universitetshospital

Dear authors,

I appreciate the opportunity to evaluate your manuscript titled "The reporting quality of published reviews of commercial and publicly available mobile health apps (mHealth app reviews): A scoping review protocol." While the study holds promise, I would like to address certain concerns. In my opinion, it would be advantageous for this protocol to be an update of your previous work (reference 28), and I hope these suggestions will help in refining your protocol.

We appreciate the time you have taken to review our protocol.

As stated on page 6, lines 34-35, this work will build on the previous review by Grainger et al. The authors emphasised a need for a reporting standard for health app-focused reviews. Our scoping review is the next step toward developing guidance to improve the reporting of mHealth app reviews and attempt to minimise the presence of the reporting issues identified by Grainger et al.

1- The effort to create a PRISMA extension for mHealth app evaluations is admirable. Nevertheless, it is necessary to provide a more explicit explanation for this objective. Clarifying the reasoning, especially considering the already available guidelines for reporting app technologies, would assist readers in comprehending the need for a new reporting guideline.

The authors need to clearly explain how their suggested expansion to PRISMA will directly tackle certain deficiencies in the documentation of mHealth app reviews (and not systematic literature reviews). Furthermore, the protocol should clarify the distinctive obstacles in reporting mHealth app reviews that justify a new expansion, taking into account general issues (mentioned by the authors in the paragraph 5 of the Introduction) that could be resolved without creating a new reporting guideline. In addition, the absence of quality appraisal aligning with the nature of scoping reviews present some challenges for creating an extension for PRISMA guideline.

In the Introduction (page 5, lines 35-44 and page 6 lines 1-5) we have given an overview of the existing guidelines available for reporting evaluations of digital technologies and highlighted that there are none available for reporting mHealth app reviews:

The EQUATOR Network (28) provides an array of guidelines for reporting evaluations of digital technologies, such as the CONSORT-EHEALTH checklist (an extension of the CONSORT checklist tailored for reporting randomised controlled trials of web-based and mobile health interventions (29)), and guidance on reporting evaluations of specific technologies, such as sensors (30), mHealth interventions (31), telehealth in clinical trials (32) and smartphone-delivered ecological momentary assessments (33). There are also several extensions of the Preferred Reporting Items for Systematic Reviews and Meta-Analyses (PRISMA) reporting guidelines (34) available for different types of literature reviews.

In contrast, no reporting guidelines exist for mHealth app reviews, and we are not aware of any currently in development. The need for a reporting standard for health app-focused reviews was emphasised in a scoping review published in 2020 (35). The authors reviewed 26 mHealth app reviews published between 2012 and 2018 and found issues in their reporting. For example, the date of the searches was sometimes unclear (38%, 10/26) or absent (15%, 4/26). The number of reviewers involved was also poorly reported in 58% of reviews, and in 83% it was unclear whether screening was performed independently. Another important finding was the absence of clinical recommendations or reporting on clinical efficacy, found in 77% of the included reviews. Adhering to reporting guidelines may help to minimise the presence of these inconsistencies in reporting, and ensure that standardised information is presented.

We have added a sentence to page 6, lines 3-5 which states that the issues in reporting identified by Grainger et al. may be addressed by adhering to a reporting guideline:

Adhering to reporting guidelines may help to minimise the presence of these inconsistencies in reporting, and ensure that standardised information is presented.

2- I hold a different viewpoint from the authors when it comes to the broad range of health topics (smoking cessation, mental health, strength training, and ...). I believe that this wide scope can make it difficult to draw specific and actionable conclusions. To overcome this challenge, I suggest adopting a more targeted approach that ensures practical and applicable outcomes.

We appreciated this insight but respectfully disagree with your point. We are not trying to make conclusions specific to the health topics reported on in the reviews. Instead, we are focussing on the reporting of mHealth app reviews in general. Similar to the PRISMA extensions for scoping reviews (PRISMA-Scr), protocols (PRISMA-P) and Abstracts, we intend for our guidance to be relevant for all health topics that use the same method.

3- I find it confusing that the authors' aim is to create an extension for PRISMA, which is designed for systematic reviews, and yet they exclude studies literature reviews. If the intention is to develop an extension for reporting systematic reviews, it would be necessary to identify the problems with reporting in literature reviews.

The aim is to develop an extension for systematic reviews of commercial and publicly available mHealth apps, rather than systematic literature reviews (for which PRISMA can already be used). This is because there are clear differences between the two types of review that require specific guidance, hence the need to develop the extension. This difference is explained on page 5, lines 22-33. We are therefore only including literature that reports on systematic mHealth app reviews.

While mHealth app reviews share features with traditional systematic literature reviews, they differ substantially in their methods and reporting (23) due to the review being of commercial and publicly available products on app stores, instead of published literature. Examples of traditional systematic reviews of literature describing apps include those on monitoring and managing mental health symptoms (24) and self-managing pregnancy (25). While literature reviews can tell us about the effectiveness of apps which have been evaluated and the results published, they do not provide a comprehensive overview of all apps that are commercially or publicly available for use by patients, healthcare professionals and the public. mHealth app reviews also differ as there are no formal requirements for the protocol to be registered, searches take place on app libraries, screening often takes place on Excel (rather than using specifically designed tools like Rayyan (26) or Covidence (27)), and they are more challenging to replicate, as apps may emerge, disappear or be updated between searches (23).

This method aligns with previous approaches to developing PRISMA extensions, for example when developing the PRISMA-Equity extension, the authors reviewed literature related to the topic only (i.e., equity).

Welch V, Petticrew M, Tugwell P, Moher D, O'Neill J, Waters E, et al. (2012) PRISMA-Equity 2012 Extension: Reporting Guidelines for Systematic Reviews with a Focus on Health Equity. *PLoS Med* 9(10): e1001333. [doi:10.1371/journal.pmed.1001333](https://doi.org/10.1371/journal.pmed.1001333)

4- In addition, it is important to note that the title states “The reporting quality of ...”, while the

protocol explains that the review does not actually include assessing quality because there are no specific tools available for reviewing mHealth apps. This potential inconsistency needs to be resolved to create a more coherent presentation.

We apologise for the confusion. In the context of our protocol as well as more generally, reporting quality refers to the quality of the reporting, such as whether (or not) items are reported clearly by the authors. Quality appraisals differ as they are used to determine the quality of the methodology used (e.g., whether the sample is representative or whether randomisation has been performed appropriately in RCTs).

We have clarified this on page 12, lines 13-14:

It has been suggested that some scoping reviews should also include quality assessments of the methodology used in the articles (44).

5- I noticed a significant dependence on the authors' own earlier research in the references. I suggest expanding the review of the literature to encompass a wider array of pertinent studies, guaranteeing a thorough and impartial comprehension of the existing terrain.

We have cited further examples of existing mHealth app reviews including the following:

- Payne HE, Moxley VB, MacDonald E. Health Behavior Theory in Physical Activity Game Apps: A Content Analysis. *JMIR Serious Games*. 2015;13;3(2):e4.
- Talwar D, Yeh YL, Chen WJ, Chen LS. Characteristics and quality of genetics and genomics mobile apps: a systematic review. *Eur J Hum Genet*. 2019;27(6):833-840.
- Chen, J.; Chu, J.; Marsh, S.; Shi, T.; Bullen, C. Smartphone App-Based Interventions to Support Smoking Cessation in Smokers with Mental Health Conditions: A Systematic Review. *Psych* 2023; 5, 1077-1100.
- Kheirinejad S, Visuri A, Suryanarayana SA, Hosio S. Exploring mHealth applications for self-management of chronic low back pain: A survey of features and benefits. *Heliyon*. 2023; 9(6):e16586.
- Devan H, Farmery D, Peebles L, Grainger R. Evaluation of Self-Management Support Functions in Apps for People With Persistent Pain: Systematic Review. *JMIR Mhealth Uhealth*. 2019; 7(2):e13080.
- Martín-Martín J, Muro-Culebras A, Roldán-Jiménez C, Escriche-Escuder A, De-Torres I, González-Sánchez M, Ruiz-Muñoz M, Mayoral-Cleries F, Biró A, Tang W, Nikolova B, Salvatore A, Cuesta-Vargas A. Evaluation of Android and Apple Store Depression Applications Based on Mobile Application Rating Scale. *Int J Environ Res Public Health*. 2021;18(23):12505.
- Tsai Z, Kiss A, Nadeem S, et al. Evaluating the effectiveness and quality of mobile applications for perinatal depression and anxiety: A systematic review and meta-analysis. *Journal of Affective Disorders*. 2022;296:443-453.

The section on page 5, lines 14-17 now reads:

Many mHealth app reviews have now been published, for varied topics including genetics (6, 7), patient-reported outcomes in oncology (8), mental health (9, 10), rheumatoid arthritis (11), strength training (12), menopause (13), exercise (14, 15), hand hygiene (16), atrial fibrillation (17), pain (18-20) and smoking cessation (21, 22).